# Chemical Composition of Essential Oil from Flowers of Five Fragrant *Dendrobium* (Orchidaceae)

**DOI:** 10.3390/plants10081718

**Published:** 2021-08-20

**Authors:** Francesco Saverio Robustelli della Cuna, Jacopo Calevo, Miriam Bazzicalupo, Cristina Sottani, Elena Grignani, Stefania Preda

**Affiliations:** 1DDS-Department of Drug Sciences, University of Pavia, 27100 Pavia, Italy; stefania.preda@unipv.it; 2Casimiro Mondino National Neurological Institute, 27100 Pavia, Italy; 3DBIOS-Department of Life Sciences and Systems Biology, University of Torino, 10125 Torino, Italy; 4DISTAV-Department of Earth, Environment and Life Sciences, University of Genova, 16132 Genova, Italy; miriam.bazzicalupo@gmail.com; 5Environmental Research Center, ICS MAUGERI SPA SB, Institute of Pavia, IRCCS, 27100 Pavia, Italy; cristina.sottani@icsmaugeri.it (C.S.); elena.grignani@icsmaugeri.it (E.G.)

**Keywords:** *Dendrobium*, essential oil, steam distillation, mass spectrometry, pollinator

## Abstract

A detailed chemical composition of *Dendrobium* essential oil has been only reported for a few main species. This article is the first to evaluate the essential oil composition, obtained by steam distillation, of five Indian *Dendrobium* species: *Dendrobium chrysotoxum* Lindl., *Dendrobium harveyanum* Rchb.f., and *Dendrobium wardianum* R.Warner (section *Dendrobium*), *Dendrobium amabile* (Lour.) O’Brien, and *Dendrobium chrysanthum* Wall. ex Lindl. (section *Densiflora*). We investigate fresh flower essential oil obtained by steam distillation, by GC/FID and GC/MS. Several compounds are identified, with a peculiar distribution in the species: Saturated hydrocarbons (range 2.19–80.20%), organic acids (range 0.45–46.80%), esters (range 1.03–49.33%), and alcohols (range 0.12–22.81%). Organic acids are detected in higher concentrations in *D. chrysantum*, *D. wardianum*, and *D. harveyanum* (46.80%, 26.89%, and 7.84%, respectively). This class is represented by palmitic acid (13.52%, 5.76, and 7.52%) linoleic acid (*D. wardianum* 17.54%), and (*Z*)-11-hexadecenoic acid (*D. chrysantum* 29.22%). Esters are detected especially in species from section *Dendrobium*, with ethyl linolenate, methyl linoleate, ethyl oleate, and ethyl palmitate as the most abundant compounds. Alcohols are present in higher concentrations in *D. chrysantum* (2.4-di-tert-butylphenol, 22.81%), *D. chrysotoxum* (1-octanol, and 2-phenylethanol, 2.80% and 2.36%), and *D. wardianum* (2-phenylethanol, 4.65%). Coumarin (95.59%) is the dominant compound in *D. amabile* (section *Densiflora*) and detected in lower concentrations (range 0.19–0.54%) in other samples. These volatile compounds may represent a particular feature of these plant species, playing a critical role in interacting with pollinators.

## 1. Introduction

The Orchidaceae family, with its huge number of species that evolved different pollination systems, is known for the variety and complexity of its floral scents, which according to Kaiser (1993), could potentially cover all the spectrum of fragrances occurring in nature [1]. Floral scent, which derives from the composition of volatile organic compounds emitted by the flowers’ tissues (floral VOCs), is fundamental for the defense against pathogens/herbivores and pollinator responses [2]. This trait, together with other characteristics of flowers, such as the color, the presence of nectar, and other peculiarities of the reproductive portions, contributes indeed to defining pollination syndromes [3]. The genus *Dendrobium* Sw., 1799 (Epidendroideae; Dendrobiinae), which accounts for about 1100 species distributed in Pacific Islands, Asia, and Australia, is one of the largest of the family [4]. As potted and cut flowers, *Dendrobium* species and hybrids are of great economic interest, being at the top ten among the most commercially traded orchid taxa [5]; several species are also grown and sold for medicinal purposes [6,7]. A large number of taxa, the great morphological diversity, and the wide distribution range have contributed to taxonomic ambiguities that are currently under debate [4,8,9]. In the phylogenetic revision of the genus, Takamiya et al. (2014) considered the presence of papillae on the flower’s lip in entities belonging to different clades. They demonstrated that this character evolved as an adaptation to bee pollination by *Dendrobium* species [4]. As stated in previous studies, bee-pollinated orchid flowers exhibit papillose carpets, identified as osmophores, structures of accumulation of substances responsible for floral fragrances [10,11]. Takamiya et al. (2014) recorded odor-producing cells in all species of Section *Densiflora* and the majority of the Section *Dendrobium*, thus hypothesizing that this character has probably been acquired after the divergence between the Asian and the Australasian Superclades [4]. Despite the great number of studies aimed at optimizing in vitro propagation protocols (i.e., Marting and Madassery, 2006; Teixera da Silva et al., 2015; Calevo et al. 2020; and references therein) [12,13,14], and at characterizing anatomical and chemical traits (Carlsward et al., 1997; Xu et al., 2013; Devadas et al., 2016 and references therein) [15,16,17], the genus *Dendrobium* has been little investigated from the point of view of the reproductive biology, and even less is known about floral volatilome [18]. To the best of our knowledge, only a few authors had carried out characterizations of floral volatiles from *Dendrobium* species. Flath and Ohinata (1982) investigated the VOCs of *D. superbum* Rchb. f. (syn. *D. anosmum* Lindl.), which is pollinated by the melon fly (*Dacus cucurbitae*), finding a significant amount of 4-phenylbutan-2-one, whose structure is closely related to another known fly attractant [19]. Brodmann et al. (2009) worked on *D. sinense* Tang and F.T.Wang and reported that this species emits (*Z*)-11-eicosen-1-ol (a molecule present in the alarm pheromone of honeybees) to attract hornets for pollination [20]. Silva et al. (2015) recognized terpenes as the most abundant class of compounds in the floral volatiles of *D. nobile* Lindl. [21]. Julsrigival et al. (2013) found a prevalence of 2-pentadecanone in *D. parishii* Rchb.f. [22]. Robustelli della Cuna et al. (2017), instead, compared the essential oil of different portions of *D. moschatum* (Buch.-Ham.) Sw., including the inflorescence: They observed differences among the volatile compositions, and then hypothesized that compounds like ketones or long-chain methyl and ethyl esters play a role as pollinator attractants [23]. The few reports dedicated to reproductive biology have stated that there are various ways for which *Dendrobium* species attract pollinators: There are cases of shelter mimicry [24,25,26,27,28], nectar rewarding [18], chemical and visual attraction [29], rest and mating place offering, or generalized food deception strategies like a simulation of other co-flowering species occurring in the same habitat [30]. In this work, we aimed to characterize and compare the floral volatiles of five *Dendrobiums* belonging to sections *Dendrobium* and *Densiflora* of the Asian Superclade [4,9]. In particular, we characterized the volatile fractions of the inflorescences of *D. chrysanthum* Wall. ex Lindl. (Figure 1A), *D. harveyanum* Rchb. f. (Figure 1B) and *D. wardianum* R.Warner (Figure 1C) from section *Dendrobium*, Core subclade of Clade A, and *D. chrysotoxum* Lindl. (Figure 1D) and *D. amabile* (Lour.) O’Brien (Figure 1E) from Clade A and C, respectively, of section *Densiflora* (according to Takamiya et al. 2014) [4].

## 2. Results

The yields of *D. amabile*, *D. chrysanthum*, *D. chrysotoxum*, *D. harveyanum*, and *D. wardianum* essential oils obtained by steam distillation from fresh flowers were evaluated as 0.09%, 0.34%, 0.33%, 0.39%, and 0.33% (weight/dry weight basis), respectively. Table 1 shows the results of qualitative and quantitative oil analyses on the Elite-5MS column. The compounds are listed in order of their elution and are reported as percentages of the total essential oil. Differences in the qualitative and quantitative compositions of the obtained essential oils have been observed. As shown in the Venn’s diagram (Figure 2), only palmitic acid was shared by all five taxa. On the other hand, 30 compounds were uniquely identified in *D. chrysotoxum*, and nine, eight, four, and three in *D. wardianum*, *D. harveyanum*, *D. chrysanthum*, and *D. amabile*, respectively. Furthermore, 21 compounds were found shared by *D. chrysotoxum* and *D. wardianum*. Below, the qualitative and quantitative description of essential oils for each taxon. The Pie chart (Figure 3) shows that the essential oils were different depending on the different species: It can be observed that the main constituents were compounds belonging to saturated hydrocarbons, acids, esters, coumarin, and alcohol classes.

*Dendrobium amabile*: The dominant compound was coumarin, accounting for 95.59% of the total essential oil. Of its derivatives, 3,4-dihydrocoumarin has been detected but in lower amounts (0.10%). The second-largest class (2.19%) is represented by saturated hydrocarbons, particularly docosane (1.94%) and heneicosane (0.25%). Aldehydes (0.88%) are represented by (*E,Z*)-2,4-decadienal and (*E,E*)-2,4-decadienal (0.72 and 0.16%). Unsaturated hydrocarbons are dominated by 10-heneicosene (0.43%).

*Dendrobium chrysanthum*: The main bulk of constituents is represented by acids accounting for 46.80% of the total essential oil, from which (*Z*)-11-hexadecenoic acid (29.22%), palmitic acid (13.52%), and (*Z*)-9-hexadecenoic acid (4.06%) are the most abundant compounds. The second-largest class is featured by saturated hydrocarbons (26.55%) from which docosane (17.53%), pentacosane (6.40%), and tetracosane (2.07%) are the most abundant compounds. Alcohols (22.81%) are dominated by 2,4-di-tert-butylphenol (22.81%). Unsaturated hydrocarbons (1.72%) are represented by 9-heptacosene (1.15%) and 1-tetradecene (0.57%). Esters are represented by methyl linoleate (1.03%).

*Dendrobium chrysotoxum*: The main bulk of constituents is represented by esters (46.59%), from which ethyl linolenate (26.98%), methyl linoleate (7.48%), ethyl oleate (5.39%), ethyl palmitate (3.05%), and 9-oxo-nonanoic acid, ethyl ester (9.28%) are the most abundant compounds. The second-largest class is represented by saturated hydrocarbons, accounting for 22.84% of the total essential oil, from which heneicosane (10.01%), tricosane (9.33%), and docosane (1.66%) are the most abundant compounds. Oxygenated terpenes (8.31%) are dominated by *trans*-verbenol (4.60%), followed by terpinen-4-ol (1.53%) and cis-verbenol (0.92%). Alcohols, accounting for 7.97% of the total essential oil, are featured by 1-octanol (2.80%), 2-phenylethanol (2.36%), and α-phellandren-8-ol (2.15%). Aldehydes (3.15%) are represented by phenylacetaldehyde (0.84%), hexanal (0.73%), (*E,Z*)-2,4-decadienal (0.48%) and (*E,E*)-2,4-decadienal (0.40%). Terpenes (2.04%) are featured by γ-terpinene (0.76%) and neocembrene (0.52%).

*Dendrobium harveyanum:* The main bulk of constituents is represented by saturated hydrocarbons, accounting for 80.20% of the total essential oil, from which eicosane (40.42%), docosane (26.82%), pentacosane (6.53%) heneicosane (2.92%), and hexacosane (2.46%) are the most abundant compounds. The second-largest class is characterized by acids accounting for 7.84% of the total essential oil. The dominant compound of this class appears to be palmitic acid (7.52%). Aldehydes (1.62%) are represented by (*E,Z*)-2,4-decadienal (0.88%) followed by (*E,E*)-2,4-decadienal (0.39%). 

*Dendrobium wardianum*: The main bulk of constituents is represented by esters (49.33%) from which ethyl linolenate (32.24%), methyl linoleate (13.17%), ethyl palmitate (0.99%), phenylacetic acid ethyl ester (0.72%), ethyl oleate (0.72%) and ethyl cinnamate (0.55%) are the most abundant compounds. The second-largest class is characterized by acids accounting for 26.89% of the total essential oil, from which linoleic acid (17.54%), palmitic acid (5.76%), and myristic acid (3.59%) are the most representative compounds. Alcohols, accounting for 7.02% of the total essential oil are featured by 2-phenylethanol (4.65%), octadecan-1-ol (0.60%), 4-vinylphenol (0.52%), benzyl alcohol (0.52%) and 2-methoxy-4-vinyl-phenol (0.24%). Terpenes (5.73%) are characterized by neocembrene (3.07%), 9-epi-(*E*)-caryophyllene (1.32%) and β-selinene (1.30%). Saturated hydrocarbons, accounting for 2.20% of the total essential oil, are represented by heneicosane (1.66%) and heptadecane (0.54%). Aldehydes (1.20%) are featured by (*E,Z*)-2,4-decadienal and (*E,E*)-2,4-decadienal (0.48 and 0.39%).

## 3. Discussion

Little is known about the pollinators of the studied species, but as argued by Dobson (2006) and Witjes et al. (2011), it is possible to reconstruct the pollinator community behind a certain species by analyzing the volatile composition of flowers [33,34]. While research is still needed to identify pollinators, our analyses constitute a first contribution for the study of compounds possibly involved in plant-animal interactions. However, other functions of floral volatiles, that may play a crucial role in herbivory avoidance and as defensive molecules against pathogens, cannot be excluded [35,36]. Differences in the floral scents of related taxa could play a role in reproductive isolation by influencing pollinator’s behavior and choices [37,38,39,40]. Indeed, in some cases, a simple change in the amount of one floral VOC has been linked with strong reproductive isolation, as seen in *Silene dioica* (L.) Clairv. and *S. latifolia* Poir. [41]. However, this ethological type of isolation seems to be more or less pivotal depending on the specialization of both the plants and pollinators considered, highlighting the need to carry out additional detailed behavioral experiments to understand plant-pollinator interactions [3].

In this work, the relative composition in floral VOCs of the five *Dendrobium* species was qualitatively studied. The highest number of species-specific compounds were recorded for entities from section *Dendrobium*. Palmitic acid was the only compound shared by all the five taxa examined. This molecule is frequently found in the volatilome of several plant species (Orchidaceae included) [23,35,42], and also in other organisms; we observed that it was relatively abundant in *D. chrysanthum* (13%), followed by *D. harveyanum* (7.52%) and *D. wardianum* (5.76%), while in the remaining two species it was less represented.

The scent recognized for both *D. chrysotoxum* and *D. wardianum* could be due to the high presence of esters in floral VOCs that we detected during our analyses. Esters are produced by the reaction of alcohols with organic acids; they typically have fruity smells and are indeed among the molecules responsible for the odors of many fruits [43]. High content of volatile esters has been linked with the strong flavor of the “snow chrysanthemum” cultivar of *Coreopsis* by Kim et al. (2020) [44]. In *D. moschatum*, a putative role as semiochemicals involved in pollinator attraction has been hypothesized for methyl and ethyl esters by Robustelli della Cuna et al. (2017) [23]. According to da Silva et al. (1999) and Cseke et al. (2007), terpenes are more abundant in flower VOCs of species pollinated by food-seeking bees [45,46]. As shown in Table 1, *D. wardianum* had the highest level (5.73%) of terpenes in the essential oil, followed by *D. chrysotoxum* (2.04%), but this class of compounds was not the predominant one in these two species. Conversely, oxygenated terpenes have been detected only in *D. chrysotoxum* (8.31%), while they were present in lower percentages in *D. harveyanum* and *D. chrysanthum*. Therefore, due to their ester and terpenoid contents, and considering similar results obtained by Flath and Ohinata (1982) for *D. superbum*, we cannot exclude that *D. chrysotoxum* and *D. wardianum* could rely on the action of frugivorous flies or bees, or other animals for their pollination [19].

It is noteworthy that the VOCs spectrum of *D. amabile*, a scented orchid, was almost entirely dominated by coumarin, a compound having a sweet smell that resembles vanilla. On the contrary, this compound was present only in very small percentages in all the other *Dendrobiums* considered. As previously stated by Robustelli della Cuna et al. 2017, coumarin was abundant, although less represented in respect to *D. amabile*, also in VOCs from inflorescences and leaves of *D. moschatum* [23]. In this species, authors hypothesized a phytoalexin-like defensive role for coumarin. In the future, a possible role of coumarin in plant-pollinator interactions should be investigated. Interestingly, *D. chrysanthum* showed a distinctive floral volatile composition compared to the other species. Indeed, this entity displayed the highest amounts of acids (accounting for 46.8% of the total essential oil), together with a good representation of alcohols (22.8%) if compared to the other species considered. Among acids, the most representative one (29.2%) was *Z*-11-Hexadecenoic acid, a known sex pheromone in moths [47]. Considering the relatively high content of this compound, we can again hypothesize its possible role as pollinator (putatively, moth) attractant. Concerning alcohols, 2,4-di-tert-butylphenol was relatively abundant in *D. chrysanthum*. This molecule was also present in traces in *D. amabile*. Zhang et al. (2017) and Huang et al. (2018) recorded the occurrence of this alcohol in flowers of *D. moniliforme* (L.) Sw. and rhizomes of *Gastrodia elata* Blume, respectively [48,49]. This compound, known for its toxicity, exerts several bioactivities and has insecticidal, nematicidal, antibacterial, and antifungal properties (Zhao et al., 2020 and references therein) [50]. Therefore, a defensive role for 2,4-di-tert-butylphenol in *D. chrysanthum* cannot be excluded. Finally, it is interesting to notice that among the *Dendrobium* and *Densiflora* sections, three self-incompatible species, *D. amabile* and *D. harveyanum*, and *D. chrysanthum*, respectively, showed a reduced spectrum of volatiles [51,52]. It is tempting to hypothesize that this has a role in pollination biology; indeed, discouraging pollinators from pollinating more flowers of the same plant and inducing pollinators to visit different individuals, would result in a higher fruit set.

## 4. Conclusions

In conclusion, this is the first study reporting the floral volatile components of *D. amabile, D. chrysanthum, D. chrysotoxum, D. harveyanum*, and *D. wardianum*. Our results can put the basis for the investigation of *Dendrobium*’s pollination biology and plant-herbivore interactions, but further studies to find the pollinators and understand their behaviors are required for deciphering the role of the compounds detected in these five *Dendrobium* species. Considering the present results, studies on the fingerprint of the essential oils of other *Dendrobium* sections (i.e., *Calcarifera, Crumenata, Fugacia, Latouria*, and *Spatulata*) are in progress in our lab.

## 5. Materials and Methods

### 5.1. Plant Material

All examined species were provided by specialized sellers, in details: *D. wardianum* (Buchanan-Hamilton) Swartz, *D. chrysotoxum* (Lindley) and *D. harveyanum* (Rchb.f.) from Orchid’s and more, (Ismaning, Germany), *D. amabile* (O’Brien) and *D. chrysanthum* (Wallich ex Lindley) from Kopf Orchideen und Floristik, (Deggendorf, Germany). Plants were identified according to Dressler (2016) [53], and cultivated under intermediate greenhouse conditions at the University of Turin, Italy, for two years before analyses. Plants were grown in intermediate conditions in a greenhouse during winter months and outside from April to October using a bark well-drained potting medium. Samples were collected at the flowering stage and stored at −20 °C until extraction. Before extraction, the flowers were brought back to room temperature and subjected to steam distillation.

### 5.2. Isolation of the Essential Oil

Flowers (*D. amabile* 25 g, *D. chrysanthum* 5.53 g, *D. chrysotoxum* 5.87 g, *D. harveyanum* 6.14 g, *D. wardianum* 6.26 g), to which octyl octanoate (98%, Sigma-Aldrich, Inc., St. Louis, MO, USA) was added as internal standard, were steam distilled with odor-free water for 3 h. The distillate was extracted with methylene chloride (3 × 100 mL) (Merck, Darmstadt, Germany), dried over anhydrous sodium sulfate (Sigma-Aldrich, Inc., St. Louis, MO, USA), and concentrated at first with a rotary evaporator and subsequently using a gentle stream of N_2_ for successive GC/FID and GC/MS analyses [23,36].

### 5.3. GC-FID Analysis

The analyses were carried out using a Hewlett Packard model 5980 GC, equipped with Elite-5MS (5% phenyl methyl polysiloxane) capillary column of (30 m × 0.32 mm i.d.) and film 0.32 μm thick. The carrier gas was He at a flow of 1 mL/min. One μL aliquots of essential oil were manually injected in splitless mode. The oven temperature program included an initial isotherm of 40 °C for 5 min, followed by a temperature ramp to 260 °C at 4 °C/min, and a final isotherm at this temperature for 10 min. Injector and detector temperatures were set at 250 and 280 °C, respectively. The relative amount of each component was calculated based on the corresponding FID peak area without response factor correction.

### 5.4. GC-MS Analysis

The analyses were carried out using a GC Model 6890 N, coupled to a benchtop MS Agilent 5973 Network, equipped with the same capillary column and following the same chromatographic conditions used for the GC/FID analyses. The carrier gas was He at a constant flow of 1.0 mL/min. The essential oils were diluted before analysis, and 1.0 µL was manually injected into the GC system with a split ratio of 30:1. The ion source temperature was set at 200 °C, while the transfer line was at 300 °C. The acquisition range was 40–500 amu in electron-impact (EI) positive ionization mode using an ionization voltage of 70 eV.

### 5.5. Identification and Quantification of the Essential Oil Components

The identification of the volatile oil components was performed by their retention indices (RI) and their mass spectra [31], and by comparison with a NIST database mass spectral library, as well as with literature data [32,54]. Retention indices were calculated by Elite-5MS capillary columns using an n-alkane series (C_6_–C_35_) (Sigma-Aldrich, Inc., St. Louis, MO, USA) under the same GC conditions as for samples. The relative amount of each component of the oil was expressed as percent peak area relative to total peak area from GC/FID analyses of the whole extracts. The quantitative data were obtained from GC/FID analyses by an internal standard method and assuming an equal response factor for all detected compounds.

## Figures and Tables

**Figure 1 plants-10-01718-f001:**
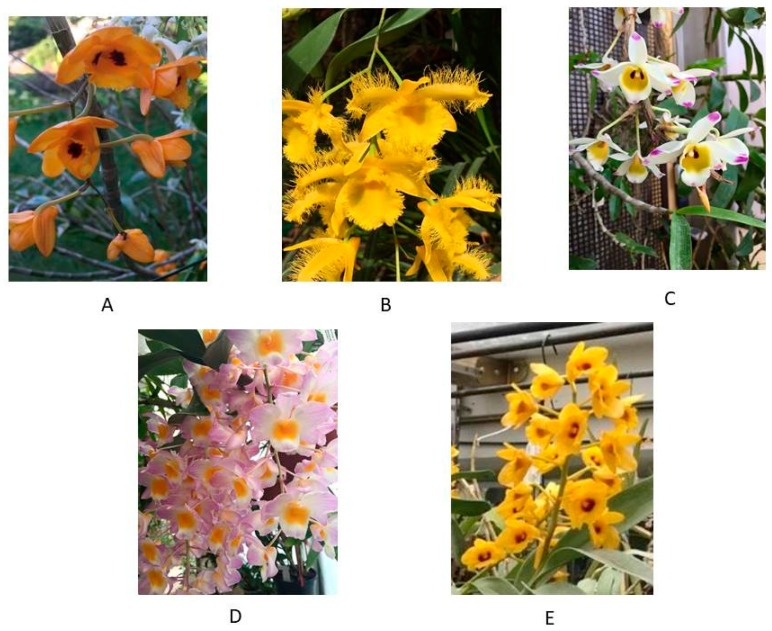
*Dendrobium chrysanthum* (**A**), *D. harveyanum* (**B**), *D. wardianum* (**C**), *D. amabile* (**D**), and *D. chrysotoxum* (**E**), greenhouse-grown plants cultivated in Turin (Italy).

**Figure 2 plants-10-01718-f002:**
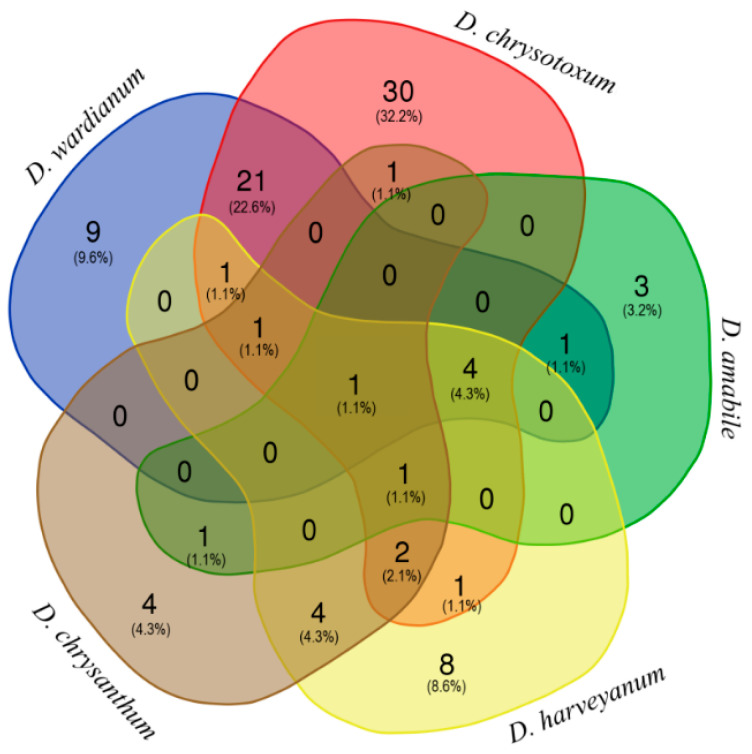
Venn’s diagram shows both the number of compounds shared and unshared/peculiar among the five *Dendrobium* species. Percentages are referred to the total number of compounds found, not to the relative abundance.

**Figure 3 plants-10-01718-f003:**
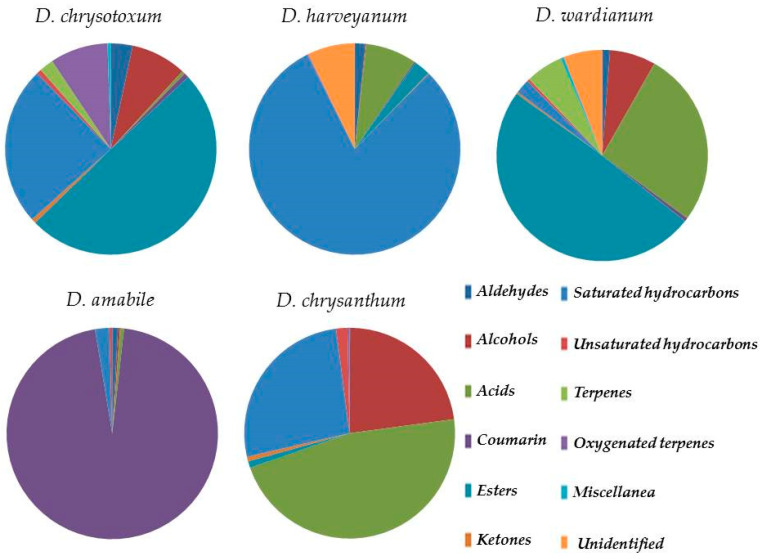
Pie chart of distribution of the classes.

**Table 1 plants-10-01718-t001:** Essential oils composition of inflorescences from the five *Dendrobium* species.

Compound ^a^	RI ^b^	RI ^c^	Section *Dendrobium*			Section *Densiflora*		
			*D. chrysotoxum*	*D. harvejanum*	*D. wardianum*	*D. amabile*	*D. chrysanthum*	Identification ^d^
			%	%	%	%	%	
Octane	800	800	-	0.15	-	-	-	RI, NIST
Hexanal	802	801	0.73	0.06	0.02	-	-	RI, NIST
2-hexanol	804	808	-	0.12	-	-	-	RI, NIST
Diacetone alcohol	841	841	-	-	-	-	0.68	RI, NIST
α-pinene	939	931	0.21	-	-	-	-	MS, NIST
Benzaldehyde	960	958	0.14	-	-	-	-	RI, NIST
β-pinene	979	973	0.03	-	-	-	-	MS, NIST
Caproic acid	1005	1003	0.06	-	-	-	-	RI, NIST
α-terpinene	1017	1015	0.10	-	-	-	-	RI, NIST
*o*-Cymene	1026	1023	0.09	-	-	-	-	RI, NIST
Limonene	1029	1027	0.17	-	-	-	-	RI, NIST
Benzyl alchol	1032	1035	0.21	-	0.52	-	-	RI, NIST
β-Isophorone	1042	1041	0.51	-		-	-	RI, NIST
Phenylacetaldehyde	1042	1043	0.84	-	0.06	-	-	RI, NIST
2-octenal	1056	1058	-	0.13	-	-	0.06	RI, NIST
γ-Terpinene	1060	1059	0.76	-	0.04	-	-	RI, NIST
Unidentified	-	1065	-	-	2.89	-	-	-
*cis*-sabinene hydrate	1070	1067	0.27	-	-	-	-	MS, NIST
dihydromyrcenol	1073	1073	-	0.04	-	-	0.06	RI, NIST
1-octanol	1070	1074	2.80	-	0.41	-	-	MS, NIST
*trans*-sabinene hydrate	1098	1098	0.20	-	-	-	-	RI, NIST
Linalool	1097	1101	0.34	0.08	-	-	-	MS, NIST
Nonanal	1102	1105	-	0.16	-	-	-	RI, NIST
2-phenylethanol	1107	1115	2.36	-	4.65	-	-	MS, NIST
Methyl octanoate	1127	1127	0.04	-	-	-	-	RI, NIST
*cis*-verbenol	1141	1142	0.92	-	-	-	-	RI, NIST
*trans*-verbenol	1145	1148	4.60	-	-	-	-	RI, NIST
Camphor	1150	1157	-	0.12	-	-	-	MS, NIST
Nonenal	1162	1161	0.41	-	0.17	-	-	RI, NIST
α-phellandren-8-ol	1170	1169	2.15	-	-	-	-	RI, NIST
Terpinen-4-ol	1177	1179	1.53	-	-	-	-	RI, NIST
Diethyl succinate	1182	1184	0.33	-	-	-	-	RI, NIST
*p*-cymen-8-ol	1183	1186	0.29	-	-	-	-	RI, NIST
α-terpineol	1189	1192	0.18	-	-	-	0.28	RI, NIST
Ethyl octanoate	1196	1199	0.20	-	-	-	-	RI, NIST
Decanal	1202	1206	-	-	0.04	-	-	RI, NIST
Verbenone	1205	1210	0.20	-	-	-	-	MS, NIST
2,4-nonandienal	1212	1214	-	-	0.03	-	-	RI, NIST
4-vinylphenol	1224	1221	-	-	0.52	0.08	-	RI, NIST
3-phenyl-1-propanol	1232	1231	-	-	0.08	-	-	RI, NIST
Phenylacetic acid ethyl ester	1247	1247	0.15	-	0.72	-	-	RI, NIST
Nerol	1254	1256	0.06	-	-	-	-	RI, NIST
2,4-decadienal (*E*,*E*)	1291	1295	0.40	0.39	0.39	0.16	-	RI, NIST
2-methoxy-4-vinyl-phenol	1315	1315	-	-	0.24	-	-	RI, NIST
2,4-decadienal (*E*,*Z*)	1319	1317	0.63	0.88	0.48	0.72		RI, NIST
2-nonenoic acid-γ-lactone	1345	1344	0.39	-	0.49	-	-	RI, NIST
Capric acid	1359	1359	-	0.32	-			RI, NIST
Eugenol	1367	1366	-	-	-	0.10	-	RI, NIST
1-tetradecene	1390	1393	-	0.07	-		0.57	MS, RI
3,4-dihydrocoumarin	1398	1399	-	-	-	0.10	-	RI, NIST
Coumarin	1434	1436	0.71	0.19	0.54	95.49	-	RI, NIST
9-epi-(*E*)-caryophyllene	1466	1458	-	-	1.32	-	-	MS, NIST
Ethyl-cinnammate	1467	1468	-	-	0.55	-	-	RI, NIST
2,4-di-tert-butylphenol	1494	1489		-		0.12	22.81	MS, NIST
β-selinene	1494	1489	0.25	-	1.30	-	-	MS, NIST
9-oxo-ethyl-nonanoate	1507	1510	1.28	-	-	-	-	MS, NIST
Lauric acid	1566	1568	0.23	-	-	-	-	RI, NIST
Ethyl laurate	1593	1596	0.15	-	-	-	-	RI, NIST
Unidentified	-	1658	-	5.16	-	-	-	-
Pentadecan-2-one	1667	1667	-	-	0.26	-	-	RI, NIST
Heptadecane	1700	1700	0.31	-	0.54	-	-	RI, NIST
Unidentified	-	1767	0.39	-	3.04	-	-	-
Myristic acid	1780	1776		-	3.59	-	-	MS, NIST
1-octadecene	1790	1796	0.32	-	0.41	-	-	MS, RI
Methyl pentadecanoate	1820	1828	0.04	-	-	-	-	MS, NIST
Unidentified	-	1879	5.74	-	-	-	-	-
Ethyl pentadecanoate	1890	1896	0.36	-	0.19	-	-	MS, NIST
Heptadecan-2-one	1902	1903	0.11	-		-	-	RI, NIST
Methyl palmitate	1927	1928	0.34	-	0.44	-	-	RI, NIST
*cis*-9-hexadecenoic acid	1942	1943	-	-	-	-	4.06	RI, NIST
*Z*-11-Hexadecenoic acid	1953	1953	-	-	-	-	29.22	RI, NIST
Palmitic acid	1958	1960	0.05	7.52	5.76	0.61	13.52	RI, NIST
Neocembrene	1960	1966	0.52	-	3.07	-	-	MS, NIST
Ethyl palmitate	1992	1997	3.05	-	0.99	-	-	MS, NIST
Octadecan-1-ol	2074	2071	0.17	-	0.60	-	-	MS, NIST
Eicosane	2000	2000	-	40.42	-	-	0.55	RI, NIST
Unidentified	-	2037	-	2.06	-	-		-
Methyl linoleate	2051	2068	7.48	2.50	13.17	-	1.03	MS, NIST
10-Heneicosene	2060	2073	-	-	-	0.43	-	MS, RI
Heneicosane	2100	2100	1.01	2.92	1.66	0.25	-	RI, NIST
Linoleic acid	2144	2147	0.12	-	17.54	-	-	RI, NIST
Ethyl linolenate	2169	2171	26.98	-	32.24	-	-	RI, NIST
Ethyl oleate	2179	2181	5.39	-	0.72	-	-	RI, NIST
Ethyl octadecanoate	2193	2198	0.80	-	0.31	-	-	RI, NIST
Docosane	2200	2204	1.66	26.82	-	1.94	17.53	RI, NIST
9-Triacosene	2279	2275	0.31	-	-	-	-	MS, RI
Tricosane	2300	2307	9.33	-	-	-	-	RI, NIST
Tetracosane	2400	2401	0.40	0.90	-	-	2.07	RI, NIST
9-Pentacosene	2474	2475	0.07		-	-		MS, RI
Pentacosane	2500	2501	0.95	6.53	-	-	6.40	RI, NIST
Hexacosane	2600	2600	-	2.46	-	-	-	RI, NIST
9-Eptacosene	2676	2676	-	-	-	-	1.15	MS, RI
Heptacosane	2700	2701	0.18	-	-	-	-	RI, NIST
Aldehydes			3.15	1.62	1.20	0.88	0.06	
Alcohols			7.97	0.12	7.02	0.30	22.81	
Acids			0.45	7.84	26.89	0.61	46.80	
Coumarin			0.71	0.19	0.54	95.59	-	
Esters			46.59	2.50	49.33	-	1.03	
Ketones			0.62	0.12	0.26	-	0.68	
Saturated hydrocarbons			22.84	80.20	2.20	2.19	26.55	
Unsaturated hydrocarbons			0.69	0.07	0.41	0.43	1.72	
Terpenes			2.04	-	5.73	-	-	
Oxygenated terpenes			8.31	0.11	-	-	0.34	
Miscellanea			0.48	-	0.49	-	-	
Unidentified			6.13	7.22	5.92	-	-	

^a)^ Compounds are listed in order of elution from an Elite-5 column. ^b)^ Retention Indices according to Adams [31], unless stated otherwise. ^c)^ Retention index (mean) determined on an Elite-5 column using a homologous series of *n*-hydrocarbons, ^d)^ Method of identification: MS, mass spectrum; NIST, comparison with library [32]; RI, retention indices in agreement with literature values.

## Data Availability

Not applicable.

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
