# Peer review of "Chemical Composition of Essential Oil from Flowers of Five Fragrant Dendrobium (Orchidaceae)"

_plants, 2021, doi:10.3390/plants10081718_

Round 1

Reviewer 1 Report

Reviewer comments:  plants

 Chemical Composition of Essential Oil from Flowers of Five Dendrobium (Orchidaceae) from Southern Asia

I read and evaluate the manuscript titled “Chemical Composition of Essential Oil from Flowers of Five Dendrobium (Orchidaceae) from Southern Asia ” by  della Cuna et al.,”.

The aim of this paper is not clear.

Why authors put results and discussion before material and methods.

In table 1, why authors put RI for each plant,?

The author should give a reference for the method used for extracting the essential oil used in this paper.

To compare these results with other studies, the same method of extraction should be used.

Authors don’t give conclusion for this study.

Author Response

Responses to Reviewer 1) As the title already says, the Aim of our work is the characterization of essential oil from five fragrant Dendrobium. However, during the review rounds we modified the abstract and other parts of the article to make it clearer2) The results and discussion paragraphs have been placed before materials and methods in accordance with the usual MDPI format3) The Table 1 has been already modified after the first round and now it’s ok.4) References have been provided after the review rounds; several published papers reporting works done using steam distillation are available and we referred to many of these.5) Conclusions have been moved and implemented during previous review rounds and are now ok.

Reviewer 2 Report

The manuscript is quite complete in the essential oil analysis section, but its writing and discussion are still insufficient.

  1. In abstract, should highlight the importance of research results.
  2. In able1, RI only needs to be presented once
  3. LINE 170, Palmitic acid was fatty acid synthesis biological product, that has not specificity.
  4. There should be enough statistics to explain the discussion.
  5. The conclusion is not presented in the manuscript.

Author Response

Thanks for your comments. Below are the changes made according to the indications.

Best Regards

Francesco Saverio Robustelli della Cuna

In abstract, should highlight the importance of research results: Amended

In able1, RI only needs to be presented once.: Amended

LINE 170, Palmitic acid was fatty acid synthesis biological product, that has not specificity: Amended

The conclusion is not presented in the manuscript: Amended

Reviewer 3 Report

     The current manuscript concerns with the evaluation of the essential oil composition of five Indian Dendrobium species through investigating fresh flowers essential oil obtained by steam distillation by GC/FID and GC/MS. It is well-planned study and the work is novel, interesting and can be considered for publication after minor revision.

Suggesting the authors to address these issues:

  • The authors should add the sources and purity percentage for all chemical compounds used in the current study.
  • Conclusion for the study is highly recommended.
  • Please adjust the references according to the journal’s instructions.

Author Response

  • The authors should add the sources and purity percentage for all chemical compounds used in the current study: Amended
  • Conclusion for the study is highly recommended: Amended
  • Please adjust the references according to the journal’s instructions: Amended

Reviewer 4 Report

This paper is definitely worth publishing. I have a few minor changes to suggest:

In the Abstract, lines 76-80 and 229-231, please use the complete taxonomic names of the species investigated: Dendrobium chrysotoxum Lindl., Dendrobium harveyanum Rchb.f., Dendrobium wardianum R.Warner, Dendrobium amabile (Lour.) O'Brien, and Dendrobium chrysanthum Wall. ex Lindl.

Line 59: Dendrobium superbum Rchb.f. is actually a synonym for Dendrobium anosmum Lindl. and this should be mentioned (with the complete taxonomic name).

Line 62: please use the complete name of this species: Dendrobium sinense Tang & F.T.Wang. The same for line 65: Dendrobium nobile Lindl.; line 66 Dendrobium parishii Rchb.f.; line 67 Dendrobium moschatum (Buch.-Ham.) Sw.; lines 163-164: Silene dioica (L.) Clairv.; Silene latifolia Poir. (or was it Silene latifolia (Mill.) Britten & Rendle, which is a synonym of Silene vulgaris (Moench) Garcke ?); line 174 Dendrobium officinale Kimura & Migo (actually a synonym for Dendrobium catenatum Lindl., which should preferably used, with its synonym in brackets); line 183 Dendrobium moschatum (Buch.-Ham.) Sw. (if not another D. moschatum, synonym to other names); line 212 - Dendrobium moniliforme (L.) Sw.; Gastrodia elata Blume;

Line 84: because the water content can vary among different samples, reporting of results on a dry matter basis is highly recommended.

The limitations in the identification of the chemical compounds should be acknowledged in the discussions section.

Author Response

Thanks for your comments. Below are the changes made according to the indications.

Best Regards

Francesco Saverio Robustelli della Cuna

Line 59: Dendrobium superbum Rchb.f. is actually a synonym for Dendrobium anosmum Lindl. and this should be mentioned (with the complete taxonomic name): Amended

Line 62: please use the complete name of this species: Dendrobium sinense Tang & F.T.Wang. The same for line 65: Dendrobium nobile Lindl.; line 66 Dendrobium parishii Rchb.f.; line 67 Dendrobium moschatum (Buch.-Ham.) Sw.; lines 163-164: Silene dioica (L.) Clairv.; Silene latifolia Poir. (or was it Silene latifolia (Mill.) Britten & Rendle, which is a synonym of Silene vulgaris (Moench) Garcke ?); line 174 Dendrobium officinale Kimura & Migo (actually a synonym for Dendrobium catenatum Lindl., which should preferably used, with its synonym in brackets); line 183 Dendrobium moschatum (Buch.-Ham.) Sw. (if not another D. moschatum, synonym to other names); line 212 - Dendrobium moniliforme (L.) Sw.; Gastrodia elata Blume: Amended

Line 84: because the water content can vary among different samples, reporting of results on a dry matter basis is highly recommended: Amended

Reviewer 5 Report

Dendrobium has been used for thousands of years and is now a popular health food worldwide. The main chemical components of Dendrobium are alkaloids, aromatic compounds, sesquiterpenoids and polysaccharides, with multiple biological activities, including immunomodulatory, neuroprotective and anti-tumor effects.

Orchidaceae is considered one of the well-represented flowering plant families, wordwide distributed accounting about 28.000 species. This abundance leads to great complexity of floral scents; in fact, orchids can potentially produce almost all the fragraces occurring in nature. This wide variety of floral scents is primarily due to the combination of the great number of orchid species and the evolution of pollination systems.

Pollination of flowers by animals is often influenced by a wide variety of volatile molecules. The floral scent in Orchid Species has the primary aim to attract and guide pollinators, playing a critical role both in long- and short-distance attraction.

Comments

  • The title is not appropriate, given the fact that the plants involved in the study were acquired from European producers, acclimatized and grown in Turin, Italy.
  • L 144 fig. 2 – Please add the corresponding percentages to the Venn diagram. The heat map technique should be used to intepret the data.
  • L238 Please explain the mass of the flowers used. Dry weight, fresh weight? Were the flowers thawed and/or dried before extraction? In L83 "fresh flowers", L237 "stored at -20°C".
  • L 272 There is no mention of statistical analyses performed on the data. It is difficult to compare results and make assumptions concering the relevance of variability when statistical significance is not considered.
  • L 152 Some graphical representations should be presented for better comprehension, where the data variability allows, e.g. the content of carbonyl compounds of the 5 flower types, using the total precentages presented at the end of table 1.
  • It is difficult to comment further when the study presents only one extract type analysed using only one chromatographic method, without statistical analyses.

Accepted with major revision

Author Response

Thanks for your comments. Below are the changes made according to the indications.

Best Regards

Francesco Saverio Robustelli della Cuna

L 144 fig. 2 – Please add the corresponding percentages to the Venn diagram. The heat map technique should be used to intepret the data: Amended

L238 Please explain the mass of the flowers used. Dry weight, fresh weight? Were the flowers thawed and/or dried before extraction? In L83 "fresh flowers", L237 "stored at -20°C.: Amended (see Materials and Methods/ Plant material)

L 152 Some graphical representations should be presented for better comprehension, where the data variability allows, e.g. the content of carbonyl compounds of the 5 flower types, using the total percentages presented at the end of table 1.: Amended using the pie chart (see Figure3 )

Round 2

Reviewer 5 Report

Ok. Accept in present form.

Author Response

Please find enclosed the revised version.

Best regards

Francesco Saverio Robustelli della Cuna
